# CASSAD: Chroma-Augmented Semi-Supervised Anomaly Detection for Conveyor Belt Idlers

**DOI:** 10.3390/s24237569

**Published:** 2024-11-27

**Authors:** Fahad Alharbi, Suhuai Luo, Abdullah Alsaedi, Sipei Zhao, Guang Yang

**Affiliations:** 1School of Information and Physical Sciences, The University of Newcastle, Newcastle, NSW 2308, Australia; 2Department of Information Technology, College of Computers and Information Technology, Taif University, Taif 21944, Saudi Arabia; 3College of Computer Science and Engineering, Taibah University, Madinah 41411, Saudi Arabia; 4Centre for Audio, Acoustics and Vibration, Faculty of Engineering and IT, University of Technology Sydney, Sydney, NSW 2007, Australia

**Keywords:** chroma features, semi-supervised anomaly detection, isolation forest, one-class SVM, idler fault detection, industrial machinery, conveyor systems

## Abstract

Idlers are essential to conveyor systems, as well as supporting and guiding belts to ensure production efficiency. Proper idler maintenance prevents failures, reduces downtime, cuts costs, and improves reliability. Most studies on idler fault detection rely on supervised methods, which depend on large labelled datasets for training. However, acquiring such labelled data is often challenging in industrial environments due to the rarity of faults and the labour-intensive nature of the labelling process. To address this, we propose the chroma-augmented semi-supervised anomaly detection (CASSAD) method, designed to perform effectively with limited labelled data. At the core of CASSAD is the one-class SVM (OC-SVM), a model specifically developed for anomaly detection in cases where labelled anomalies are scarce. We also compare CASSAD’s performance with other common models like the local outlier factor (LOF) and isolation forest (iForest), evaluating each with the area under the curve (AUC) to assess their ability to distinguish between normal and anomalous data. CASSAD introduces chroma features, such as chroma energy normalised statistics (CENS), the constant-Q transform (CQT), and the chroma short-time Fourier transform (STFT), enhanced through filtering to capture rich harmonic information from idler sounds. To reduce feature complexity, we utilize the mean and standard deviation (std) across chroma features. The dataset is further augmented using additive white Gaussian noise (AWGN). Testing on an industrial dataset of idler sounds, CASSAD achieved an AUC of 96% and an accuracy of 91%, surpassing a baseline autoencoder and other traditional models. These results demonstrate the model’s robustness in detecting anomalies with minimal dependence on labelled data, offering a practical solution for industries with limited labelled datasets.

## 1. Introduction

Belt conveyors are critical for transporting materials across varied and often challenging terrains. Their operation frequently involves hazardous environments where manual inspections are difficult due to restricted access and risks such as exposure to hot surfaces, moving parts, noise, heat, and dust [1]. Maintenance of these systems is further complicated by the increasing number of components, such as idlers, required to support longer conveyor belts [1,2]. Without effective maintenance strategies, the risk of failures escalates, leading to issues such as belt misalignment, increased friction, and bearing failures. These faults reduce the overall efficiency of conveyor systems, resulting in costly unplanned downtime and potential damage to other components.

Researchers have explored various fault detection methods to address these challenges using microphones, accelerometers, distributed optical fibre sensors (DOFS), and thermal imaging [3]. For example, thermal imaging combined with YOLOv4 has achieved high accuracy (93.8%) in detecting roller faults but struggles with early-stage detection [4]. Vibration-based techniques can detect a wide range of defects but require contact sensors that are difficult to install and maintain on large-scale systems [1]. Acoustic monitoring has emerged as a promising alternative, offering a non-contact, non-destructive solution for detecting subtle anomalies in sound patterns. However, industrial noise often hinders the effectiveness of acoustic methods, necessitating robust noise filtering and preprocessing to isolate critical fault signals.

Many studies on intelligent fault detection, such as those by [2,5,6,7,8], have focused on idler fault detection using supervised learning approaches. Signal processing techniques have included methods such as signal mean and standard deviation normalisation [9,10], wavelet packet decomposition (WPD) [11,12], Mel frequency cepstral coefficients (MFCC) [7], and fast Fourier transform (FFT) [13], among others [14]. Additionally, dimensionality reduction techniques like PCA, stacked autoencoders, and Pearson correlation have been applied to improve analysis efficiency [15,16]. These methods have demonstrated significant potential, particularly in environments with abundant labelled data. However, they often need help to generalise unseen faults and are limited by the challenges of acquiring extensive labelled datasets in industrial settings.

Recent advancements in anomalous sound detection (ASD) have shifted from traditional statistical approaches to machine learning models, such as autoencoders (AEs), which can model the distribution of normal sound signals and identify faults based on reconstruction errors [17,18]. A key advantage of AEs is their reliance solely on normal operational data, eliminating the need for labelled fault data and avoiding the risks associated with inducing faults artificially. On the other hand, transformer-based models have demonstrated exceptional fault detection accuracy, with variational mode decomposition (VMD) and Swin transformers, achieving over 99% diagnostic accuracy [19]. Despite their success, these approaches are computationally intensive and heavily reliant on large labelled datasets, making them less practical for real-time industrial applications.

Given the challenges of data scarcity, noise, and the need for computational efficiency, this paper introduces the chroma-augmented semi-supervised anomaly detection (CASSAD) model, a novel approach tailored to real-world conveyor systems. The CASSAD model employs one-class SVM combined with chromagram features, which have rarely been investigated in industrial acoustic anomaly detection. Existing research often uses individual chroma features, such as Chroma CQT, Chroma CENS, or Chroma STFT. In contrast, this work explores the combination of these features, leveraging their complementary strengths to enhance anomaly detection performance. Chromagram features, known for their robustness in capturing tonal variations, are particularly suitable for detecting subtle anomalies in sound patterns. However, their application in fault detection has been limited, as researchers often prioritize other features or fail to exploit the combined potential of multiple chroma variants by integrating and optimizing these features alongside noise filtering and data augmentation techniques, such as additive white Gaussian noise (AWGN), meaning that the CASSAD model offers a unique advantage in acoustic anomaly detection. Unlike supervised models that depend heavily on labelled fault datasets, CASSAD leverages only normal operational data, making it more practical for environments where faults are rare but critical to detect. The CASSAD model is compared against other anomaly detection techniques, such as local outlier factor (LOF) and isolation forest, to validate the proposed approach under varying operational conditions. Our results demonstrate that the CASSAD model improves fault detection robustness and addresses key challenges like noise interference and data scarcity. This scalability and adaptability make it a highly effective solution for dynamic industrial environments.

The paper is structured as follows: Section 2 provides an overview of existing anomaly detection models, including the autoencoder (AE), one-class SVM (OCSVM), isolation forest (iForest), and local outlier factor (LOF), highlighting their respective strengths, limitations, and applicability to fault detection in industrial settings. Section 3 introduces the chroma-augmented semi-supervised anomaly detection (CASSAD) model, explaining its methodology, feature extraction, and preprocessing steps. Section 4 outlines the experimental setup, including data collection, hyperparameter tuning, and evaluation metrics. Finally, Section 5 presents the results and discussion, evaluating the performance of the proposed model, analyzing the impact of noise filtering, comparing it with previous studies, and offering conclusions and recommendations for future research.

## 2. Background Information

This section provides an overview of several anomaly detection models. We include a detailed explanation of the autoencoder (AE) and introduce the key semi-supervised anomaly detection algorithms: one-class SVM (OCSVM), isolation forest (iForest), and local outlier factor (LOF). We emphasize OCSVM as our core model because it can define normal operation boundaries, making it ideal for identifying abnormal acoustic patterns. Isolation forest and LOF are included for comparison, offering alternative approaches based on data partitioning and density estimation, respectively. This allows us to evaluate OCSVM’s effectiveness while exploring the strengths and limitations of other models.

### 2.1. Autoencoder Architecture

An autoencoder is a neural network that compresses input data into a lower-dimensional representation (encoding) and then reconstructs it back to its original form (decoding) [20,21]. The main goal is to minimise the reconstruction error, making the autoencoder especially useful for anomaly detection. In this context, the autoencoder is trained only on normal data, and anomalies are identified based on higher reconstruction errors [22,23]. Setting an appropriate reconstruction error threshold is crucial for anomaly detection. This threshold is derived from the validation set, and any data points with reconstruction errors above this threshold are classified as anomalies.

The general architecture of the autoencoder, shown in Figure 1, consists of the following components:Encoder: Compresses input data into a lower-dimensional representation.Bottleneck: Holds the compressed input representation, capturing essential features.Decoder: Reconstructs the input from its encoded form, attempting to match the original closely.Reconstruction Loss: The difference between the original input and its reconstruction is evaluated using:-Mean squared error (MSE), which calculates the average squared difference between actual and reconstructed values [24,25,26].-Mean absolute error (MAE), which measures the average magnitude of errors, offering less sensitivity to outliers [24].-Huber loss, which balances MSE and MAE by being quadratic for small errors and linear for larger ones, making it robust to outliers [27].

The model is trained using backpropagation, continuously minimising reconstruction loss. This enables the autoencoder to serve as an effective baseline for anomaly detection by comparing reconstruction errors against the threshold.

### 2.2. Semi-Supervised Anomaly Detection Models

Semi-supervised anomaly detection techniques, such as one-class SVM (OCSVM), isolation forest (iForest), and local outlier factor (LOF), offer a practical solution when labelled data are scarce. These models work well when only a small fraction of the dataset is labelled, allowing the model to learn from labelled and unlabelled data.

Since they do not require a fully labelled dataset of anomalies, they are particularly useful in situations where abnormal instances are rare. By primarily learning the patterns of normal behaviour during training, these models become adept at spotting deviations or anomalies in the data, making them highly effective in real-world scenarios where labelling every anomaly is often impractical.

#### 2.2.1. One-Class SVM (OCSVM)

The one-class SVM algorithm is specifically designed for anomaly detection, making it well-suited for high-dimensional data and nonlinear decision boundaries [28]. It constructs a “maximum-margin hyperplane” based on normal data to distinguish between normal and abnormal instances:(1)$$\min_{\omega ,\xi ,\rho }\frac{1}{2}{∥\omega ∥}^{2}+\frac{1}{\nu n}\sum_{i=1}^{n}\xi_{i}-\rho .$$

This is subject to the following:$$\left(\omega \cdot \phi \left(x_{i}\right)\right)\geq \rho -\xi_{i},$$
where ω is the weight vector of the hyperplane, $\phi (x_{i})$ represents the feature mapping of data points, $\xi_{i}$ are slack variables, ρ is the offset from the origin, and ν is a parameter that controls sensitivity to outliers.

This model has been widely used for detecting anomalies in audio data related to machine failures [29,30].

#### 2.2.2. Isolation Forest (iForest)

Isolation forest is an efficient algorithm that detects anomalies by isolating them in a tree structure [31]. The length of the path from the root to the isolated instance indicates an anomaly:(2)$$s\left(x,n\right)=2^{-\frac{E(h(x))}{c(n)}},$$
where h(x) is the path length from the root node to the node isolating *x*, E(h(x)) is the expected path length for all points in the forest, and c(n) is a normalisation factor based on the size of the dataset. Isolation forest has proven effective for anomaly detection in various contexts, including audio anomaly detection for machine failures [30].

#### 2.2.3. Local Outlier Factor (LOF)

The local outlier factor (LOF) algorithm identifies anomalies using a density-based approach, comparing the local density of a point to that of its neighbours [32]. The LOF value is calculated as follows:(3)$$LOF\left(x\right)=\frac{1}{\left|N(x)\right|}\sum_{y\in N(x)}\frac{{\mathrm{reachability}}_{-}\mathrm{distance}(x,y)}{{\mathrm{reachability}}_{-}\mathrm{distance}(y,x)},$$
where

N(x) represents the set of neighbours of *x*, determined using the *k*-nearest neighbors (k-NN) algorithm.${\mathrm{reachability}}_{-}\mathrm{distance}(x,y)$ is defined as follows:
$${\mathrm{reachability}}_{-}\mathrm{distance}(x,y)=\max\{k-\mathrm{distance}(y),\mathrm{distance}(x,y)\}.$$Here, k-distance(y) is the distance from *y* to its *k*-th nearest neighbour, and distance(x,y) is the raw distance between points *x* and *y*.

The reachability distance normalises the raw distance by considering the density around *y*, ensuring robustness against noise and local variations. This measure compares the local density at *x* with the density of its neighbours. A significantly lower density around *x* results in a high LOF value, indicating that *x* is an outlier. In this study, LOF is applied to chroma coefficients extracted from audio clips. These chroma coefficients are aggregated using either the mean or standard deviation. As demonstrated in [33], this approach effectively identifies anomalies in sound signals by leveraging local density comparisons. The key semi-supervised anomaly detection models explored in this study, including LOF, are compared in Table 1.

## 3. The Proposed Anomaly Detection Model

The chroma-augmented semi-supervised anomaly detection (CASSAD) model is designed to detect anomalies in industrial machinery, particularly focusing on conveyor belt idlers, by leveraging advanced chroma-based preprocessing and one-class support vector machines (OCSVM). The goal is to identify subtle acoustic changes that indicate abnormal behaviour across different operational stages of the machinery. The visual breakdown of the model is depicted in Figure 2, structured into four stages labelled A–D.

The chroma-augmented semi-supervised anomaly detection (CASSAD) model is designed to detect anomalies in conveyor belt idlers using chroma-based features and one-class support vector machines (OCSVM). The methodology begins with an audio dataset from idler machines, split into a training dataset of normal operation data, augmented with additive white Gaussian noise (AWGN) for robustness, and a test dataset containing both normal and abnormal conditions (A).

In preprocessing, key chroma features, e.g., chroma CQT, chroma STFT, and chroma CENS, are extracted from the audio signals to capture harmonic content over multiple time frames (376), each represented by 12 chroma coefficients. These chroma features are smoothed with a median filter to reduce noise and stabilise temporal variations, then concatenated into a single feature set. Dimensionality is reduced by computing statistical summaries, such as the mean and standard deviation, across time frames to transform the chroma features into compact one-dimensional representations (B). The processed chroma features are then fed into an OCSVM model and trained on normal data to learn the typical operational sound patterns of the machinery. The model identifies deviations from these patterns during testing, detecting potential anomalies that indicate machine faults (C).

Finally, the model’s performance is evaluated using AUC, accuracy, precision, recall, and F1 score metrics. Raw and smoothed chroma features are compared to assess the impact of preprocessing on detection accuracy (D). This comprehensive model enhances anomaly detection capabilities, even in noisy environments.

### 3.1. Feature Extraction Chromagram

The chromagram is a method for extracting features from music audio signals, emphasising the analysis of audio pitch [36]. The basic concept of chroma features combines all spectral data related to a specific pitch class into one single value [37]. Chroma-based properties are especially beneficial in recognising the strength of each of the 12 unique musical chroma within an octave at every time frame. These properties can distinguish between audio signals’ pitch class profiles and have succeeded in classifying industrial sounds, including those from fans, sliders, and valves [38]. Our model employs chroma CENS, chroma STFT, and chroma CQT as features, utilising 12 chroma bands for analysis.These features are extracted with a consistent hop length of 512 samples, which results in 376 time frames for the given audio segment.

#### 3.1.1. Chroma STFT

Chroma STFT employs the short-time Fourier transform (STFT) to extract chroma features, as depicted in Figure 3. The chromagram STFT provides insights into pitch classification and signal structure, highlighting areas of high intensity in red and low intensity in blue, as indicated by the colour bar next to the graph. Figure 3a illustrates four processing stages: original, harmonic, filtered, and smoothed, each displaying distinct patterns. In the original stage, the raw chroma characteristics are captured; the harmonic stage isolates harmonic content; the filtered stage reduces noise; the smoothed stage further refines the prominent chroma features. The filtered and smoothed stages will be explained in Section G. These patterns tend to be uniform and consistent for normal idler sounds. In contrast, the patterns show more variability and dispersion for abnormal idler sounds.

#### 3.1.2. Chroma CQT

The chroma CQT, shown in Figure 3b, uses the technique of constant-Q transform (CQT). It modifies the STFT to have a logarithmic space between frequency bins. The central frequency is changed with a window function, which enables an increase in bin width at lower frequencies and a decrease at higher ones for easing computational load [39]. Unlike the Fourier transform, this CQT keeps the buffer size the same across all frequencies and sets fundamental frequencies geometrically. Normal idler sounds show concentrated and steady patterns, while abnormal idler sounds display broader and more erratic distributions.

#### 3.1.3. Chroma CENS

Chroma CENS, shown in Figure 3c, adds another layer of complexity by incorporating the short-term statistical analysis of chroma band energy distributions. This variation, known as chroma energy normalised statistics (CENS), is designed to be more resilient to temporal and timbre variations [40]. CENS features are created by quantising and smoothing chroma vectors, with optional downsampling. This approach is closely linked to the short-term harmonic characteristics of fundamental audio signals, capturing elements such as rhythm, intonation, musicality, note classes, and slight timing deviations. The lower spatial resolution of CENS is particularly effective for analysing the 12 pitch classes commonly recognised in Western music. In this context, normal idler sounds exhibit more uniform and well-defined chroma energy distributions, while abnormal idler sounds display less consistent and more varied energy patterns.

### 3.2. Statistical Analysis of Chroma Features

To further enhance our analysis, we compute statistical features from the chromagram, specifically focusing on chroma features’ mean and standard deviation across time frames. These statistics provide insights into the average behaviour and variability of pitch intensity within the audio signals.

Each of the chroma features, chroma CQT, chroma STFT, and chroma CENS, yields a similar output in the form of a matrix with dimensions of 12 × 376, where the first dimension (12) represents the chroma frequency bins corresponding to the 12 pitch classes (C to B), and the second dimension (376) represents the time frames derived from segmenting the audio signal based on the chosen hop length. This segmentation allows for the detailed analysis of harmonic content across time.

We compute each chroma bin’s mean and standard deviation across the time axis (376 frames) to reduce dimensionality while preserving key information. This operation condenses the time-varying chroma features into a single vector of length 12 for each feature extraction method (chroma CQT, chroma STFT, and chroma CENS). By concatenating these vectors, the final feature vector consists of 36 values, encompassing the statistical summary of all three chroma feature sets.

The mathematical formulations for the mean μ and standard deviation σ of the chroma features are calculated as follows:(4)$$\mu =\frac{1}{N}\sum_{i=1}^{N}x_{i},$$
(5)$$\sigma =\sqrt{\frac{1}{N}\sum_{i=1}^{N}{\left(x_{i}-\mu \right)}^{2}},$$
where $x_{i}$ represents the intensity values of chroma features at each time frame, and *N* is the total number of frames. These computations enable us to quantify chroma features’ central tendency and dispersion, enhancing our understanding of their distribution and aiding in the robust classification of audio signals. Figure 4 visually demonstrates the mean and standard deviation distribution for CQT, CENS, and STFT features in both normal and abnormal signals.

Principal component analysis (PCA) is an essential statistical technique for reducing data dimensions in data analysis and machine learning [41]. PCA transforms the original set of variables into a new set of variables that are not correlated with each other. These new variables, called principal components, capture the most data variability using the fewest components. The process involves calculating the eigenvalues and eigenvectors of the data’s covariance matrix, with the principal components corresponding to the eigenvectors associated with the largest eigenvalues. This transformation is particularly beneficial for visualising high-dimensional data, improving computational efficiency, and reducing noise.

### 3.3. Enhanced Chroma Feature Analysis Through Filtering and Smoothing

In our study, we use median filtering to enhance the extraction and analysis of chroma features from idlers. This approach, based on methodologies outlined in the *librosa* documentation [42] and the work of Cho et al. [43], uses nearest-neighbour smoothing to reduce noise and remove irregularities.

This processing is essential for isolating the distinct acoustic signatures of idler components, enabling a precise evaluation of their operational condition. By identifying anomalies early, it facilitates the detection of potential issues before they develop into major failures.

Mathematically, given an original chroma feature matrix C(t,f), where *t* represents time frames and *f* represents frequency bins, we apply a median filter *M* over a sliding window. This process smooths data while preserving important edges. The filtered chroma matrix, $C_{\mathrm{filtered}}\left(t,f\right)$, is computed as follows:(6)$$C_{\mathrm{filtered}}\left(t,f\right)=\mathrm{Median}\left(C(t-w:t+w,f)\right),$$
where *w* is the window size that determines the neighbouring data points considered during smoothing. By averaging the chroma features over time, this operation effectively reduces transient noise while retaining the harmonic structures necessary for accurate analysis.

Visual representations are crucial in this analysis, particularly when comparing the original chroma features with their harmonic-enhanced counterparts. Using *matplotlib*, we generate visualisations highlighting the improvements in clarity and focus achieved through harmonic processing in Figure 3, making identifying patterns in the data easier. Although this technique significantly reduces noise, some residual artifacts may still exist in the harmonic chroma features.

To further refine the chroma features, we apply a non-local filtering technique, which compares cosine similarities between neighbouring data points and aggregates them using a median filter. This process is mathematically expressed as follows:(7)$$C_{\mathrm{non}-\mathrm{local}}\left(t,f\right)=\mathrm{Min}\left(C_{\mathrm{harmonic}}\left(t,f\right),{\mathrm{NN}}_{-}\mathrm{filter}\left(C_{\mathrm{harmonic}}\left(t,f\right)\right)\right),$$
where $C_{\mathrm{harmonic}}\left(t,f\right)$ represents the harmonic-enhanced chroma feature matrix, and NN_filter refers to the nearest-neighbour filtering operation based on cosine similarity. The “Min” operation helps eliminate outliers by taking the element-wise minimum between the original harmonic chroma features and the filtered version, enhancing the robustness of the features used in the model. These improvements are visually demonstrated in Figure 3, where the final subplots highlight the enhanced clarity and focus achieved through our filtering and smoothing techniques. The refinements ensure a more accurate and reliable representation of the acoustic signatures, leading to more effective predictive maintenance and fault detection in industrial settings.

### 3.4. Data Augmentation Using Additive White Gaussian Noise

Data augmentation is crucial in audio signal processing, especially when the goal is to improve machine learning models’ robustness [44]. One frequently used strategy involves additive white Gaussian noise (AWGN) augmentation, which adds Gaussian noise into audio signals. This technique simulates real-world noisy conditions, allowing models to generalise better across various noise levels. AWGN adds Gaussian noise n(t) to the original waveform x(t), with the signal-to-noise ratio (SNR) being carefully where lower SNRs indicate more challenging conditions due to the dominance of noise, making fault detection more difficult [45] and controlled to vary between a minimum and maximum value. Adding white noise has significantly improved the accuracy of various fault detection and audio classification tasks [46,47].

Given a waveform x(t) of length *L*, Gaussian noise n(t) is added to the waveform to achieve a desired SNR, ranging between ${\mathrm{SNR}}_{\min}$ and ${\mathrm{SNR}}_{\max}$. In our case, we chose SNR values of 15 and 30.

**Normalised signal power**: The original waveform is normalised using a constant $C=2^{\left(b-1\right)}$, where *b* is the bit depth (for instance, 16 bits in this case). The normalised waveform $x_{\mathrm{norm}}\left(t\right)$ is given by the following:(8)$$x_{\mathrm{norm}}\left(t\right)=\frac{x(t)}{C}.$$

The signal power is computed as follows:(9)$$P_{\mathrm{signal}}=\frac{1}{L}\sum_{t=1}^{L}x_{\mathrm{norm}}^{2}\left(t\right).$$

**Noise power**: Noise n(t) is sampled from a Gaussian distribution n(t)∼N(0,1). The normalised noise $n_{\mathrm{norm}}\left(t\right)$ is as follows:(10)$$n_{\mathrm{norm}}\left(t\right)=\frac{n(t)}{C}.$$

The noise power is computed as follows:(11)$$P_{\mathrm{noise}}=\frac{1}{L}\sum_{t=1}^{L}n_{\mathrm{norm}}^{2}\left(t\right).$$

**SNR and covariance calculation**: A random SNR ${\mathrm{SNR}}_{\mathrm{rand}}$ is chosen within the range $\left[{\mathrm{SNR}}_{\min},{\mathrm{SNR}}_{\max}\right]$. The covariance factor σ is computed to scale the noise as follows:(12)$$\sigma =\sqrt{\frac{P_{\mathrm{signal}}}{P_{\mathrm{noise}}}\times 10^{-{\mathrm{SNR}}_{\mathrm{rand}}/10}}.$$

This factor scales the noise to match the desired SNR, simulating different environmental conditions.

**Augmented** waveforms: The augmented waveform $x_{\mathrm{aug}}\left(t\right)$ is generated by adding the scaled noise to the original waveform:(13)$$x_{\mathrm{aug}}\left(t\right)=x\left(t\right)+\sigma \cdot n\left(t\right).$$

This process is repeated four times to generate augmented versions of the waveform, increasing the variability in the dataset.

## 4. Experiments

This section describes the experimental setup, data collection process, hyperparameter selection, and evaluation metrics for implementing the proposed model for anomaly detection in conveyor belt idlers.

### 4.1. Experimental Setup

The proposed model was implemented in a Windows 11 environment, utilising Python 3.7.16 and TensorFlow 2.10.0. An NVIDIA RTX 4080 GPU, manufactured by NVIDIA Corporation, Santa Clara, CA, USA. was used for acceleration to enhance computational efficiency. The system was powered by high-performance hardware, featuring a 13th-generation Intel(R) Core(TM) i9-13900HX processor, manufactured by Intel Corporation, Santa Clara, CA, USA, with a base clock speed of 2.20 GHz, hyper-threading across 32 cores, and supported by 32 GB of RAM for optimal performance.

### 4.2. Dataset

Data were collected by driving a vehicle along a conveyor belt, with the recording device positioned outside the car window. When a loud noise was detected, the vehicle was stopped, maintaining a distance of approximately 2–4 m from the sound source. This method was carefully designed to capture a diverse range of operational scenarios, including normal operation and multiple fault stages, ensuring variability in the acoustic signals. This variability was informed by domain knowledge of the system’s behaviour, ensuring the dataset reflects realistic operating conditions. The approach is critical for the automated identification of faults in belt conveyor idlers. Permission was granted by Port Waratah Coal Services (PWCS) in New South Wales, Australia, to gather high-quality data for this research. The B&K Type 2250 Hand-Held Analyzer, manufactured by Brüel & Kjær, Nærum, Denmark [48] was used. The B&K Type 2250 is known for its precision and is widely used in industrial acoustic measurements. The focus was on capturing sounds from idler bearings, which are critical to idler functionality and often linked to faults [1]. The recordings, which captured a range of sounds from normal operation to various stages of bearing faults, were stored in the .WAV format.

The subsequent step in analysing the audio data of conveyor idlers involved categorising the data into distinct classes based on their conditions: normal, stage 1, stage 2, and stage 3. To ensure that each audio clip corresponds to a specific class and to standardise the duration for machine learning purposes, each audio recording was split into 4-s segments. Following this processing, the data were labelled as either “normal” or “abnormal”, with normal as normal and all other segments classified as “abnormal”. This categorisation is crucial for the one-class classification task as it enables the model to be trained exclusively on normal sounds, allowing for the accurate identification and detection of faults in idlers, which typically manifest as abnormal conditions in the data. Table 2 summarises the distribution of samples across different conditions.

### 4.3. Hyperparameters

The hyperparameters for each model, such as contamination levels and kernel types, are carefully selected based on empirical testing to ensure optimal performance, as shown in Table 3. The models are trained on the augmented and non-augmented datasets and evaluated using the test dataset. The results, including confusion matrices and execution times, provide insights into the best-performing model for detecting anomalies in idler machinery. Overall, the proposed methodology effectively combines chroma feature extraction with traditional machine learning models, demonstrating robust performance in detecting subtle anomalies in industrial audio data.

### 4.4. Evaluation Metrics

Several metrics were used to assess the effectiveness of the developed semi-supervised detection model for idler components. Often used in situations of binary classification, these measures comprise the accuracy, precision, recall, unusual F1-score, and area under the ROC curve (AUC). These metrics help to confirm that the created model correctly detects irregularities while keeping false alerts at a minimal level. The key metrics are defined as follows:**Accuracy**: The overall proportion of correctly classified instances, including normal and anomalous observations.
(14)$$\mathrm{Accuracy}=\frac{TP+TN}{TP+TN+FP+FN},$$
where TP denotes true positives, TN denotes true negatives, FP refers to false positives, and FN indicates false negatives.**Recall (sensitivity or true positive rate)**: The fraction of actual positives correctly identified.
(15)$$\mathrm{Recall}=\frac{TP}{TP+FN}.$$**Precision**: The ratio of true positive predictions to the total predicted positives.
(16)$$\mathrm{Precision}=\frac{TP}{TP+FP}.$$**Abnormal** F1-score: A specific F1-score is calculated for detecting abnormal instances, emphasising accurately identifying anomalies. It provides a focused evaluation of how well the model captures abnormal behaviours.
(17)$$\mathrm{Abnormal}\;F1-\mathrm{Score}=2\times \frac{\mathrm{Abnormal}\;\mathrm{Precision}\times \mathrm{Abnormal}\;\mathrm{Recall}}{\mathrm{Abnormal}\;\mathrm{Precision}+\mathrm{Abnormal}\;\mathrm{Recall}}.$$**The receiver operating characteristic (ROC) curve**: This is a visual display used for assessing the effectiveness of a binary classifier [49]. It shows the true positive rate (sensitivity) versus the false positive rate (1-specificity) at different threshold levels. The ROC curve aids in seeing how changes to decision thresholds impact balances between true positives and false positives.**Area under the ROC curve (AUC)**: The AUC is a metric to measure the model’s overall ability to differentiate between classes. It supplies one numerical value, summarising how well the model performs across all classification limits. If the AUC gets a higher score, it means better performance by the model, with 1 signifying an ideal classification and 0.5 showing no power for discrimination.

### 4.5. Threshold Selection Process

Selecting an appropriate decision threshold is critical to our anomaly detection model for conveyor belt idlers. The choice of decision boundary decides when the model labels any sample as normal or abnormal. Picking the suitable boundary can help recognise problems early and reduce false positives, which is vital for effective maintenance tasks. We follow a structured approach to optimise the selection of the best threshold, ensuring a balance between performance and accuracy. This process involves analysing the model’s decision scores and leveraging key evaluation metrics, such as the ROC curve and Youden’s J statistic, to determine the optimal cutoff point.

**Youden’s J statistic and ROC curve analysis**: We compute Youden’s J statistic for each potential threshold. This statistic helps us identify where we achieve the best trade-off between detecting true positives (faults) and minimising false positives (incorrect fault detections). The formula for Youden’s J is as follows:
(18)J=TruePositiveRate(TPR)−FalsePositiveRate(FPR).The optimal threshold maximises this statistic, ensuring a balance between sensitivity (the ability to detect faults) and specificity (the ability to avoid false positives).**Visualising the ROC curve**: The ROC curve lets us see the model’s performance at various levels by showing the true positive rate (TPR) versus the false positive rate (FPR). The AUC offers a comprehensive evaluation of how good the model is. A higher AUC means better performance. The ROC curve also helps us choose a threshold that aligns with the operational goals of high sensitivity and minimal false alarms.

#### 4.5.1. Optimal Threshold Selection

Based on the ROC curve and Youden’s J statistic, we select the threshold that best balances detecting faults and minimising false positives. Once this threshold is determined, the model classifies each sample as follows:(19)$$\mathrm{prediction}=\left\{\begin{array}{cc} 1 & \mathrm{if}\;\mathrm{the}\;\mathrm{decision}\;\mathrm{score}\geq \mathrm{threshold}\\ -1 & \mathrm{if}\;\mathrm{the}\;\mathrm{decision}\;\mathrm{score}<\mathrm{threshold} \end{array}.\right.$$

This threshold ensures that the system identifies faults early while keeping the rate of false positives low, allowing for effective and timely maintenance without unnecessary interventions.

#### 4.5.2. Threshold Validation

We validate the threshold by assessing model performance across key metrics like AUC to ensure it is effective. This involves analysing the ROC curve to confirm that the chosen threshold maximises the detection of true faults while minimising false positives. Table 4 below shows the performance of various models at their optimal ROC thresholds based on sample data selected randomly. By evaluating and validating these thresholds, we ensure that our model strikes the right balance between detecting real faults and minimising false alarms. Figure 5 shows the ROC curves for the best-performing models, highlighting their AUC values and best ROC thresholds. This allows us to maintain a high level of operational efficiency while effectively managing potential faults in conveyor belt idlers.

## 5. Results and Discussion

This section presents the proposed model’s results, beginning with baseline detection using the autoencoder. It then covers the detection outcomes of isolation forest and LOF, followed by the proposed model’s performance evaluation. Additionally, the section provides a visual analysis of the leading anomaly detection techniques, explores the impact of noise filtering, and compares the findings with previous studies, concluding with a brief discussion.

### 5.1. Baseline Detection Results Using an Autoencoder

In our study on anomaly detection with autoencoders, we utilised chroma features with dimensions of 12 × 376 (12 chroma bins and 376 timeframes) as input to the autoencoder model. We tested different loss functions to determine the most effective configuration for identifying anomalies in idlers. The training set, after augmentation, contained 512 normal samples, while the testing set included 211 abnormal and 129 normal samples. We trained the autoencoder using the Adam optimiser with a learning rate of 0.01 to optimise its performance. The model was trained for 30 epochs with a batch size of 4, as shown in Table 3, using shuffled data and a 10% validation split to track progress. After training, we used the reconstruction error from the normal samples in the training set to estimate a classification threshold, calculated as the mean reconstruction error plus one standard deviation. This threshold was then applied to the test data, classifying samples as normal or anomalous based on whether their reconstruction error was below or above the threshold. This approach allowed the autoencoder to act as a classifier, systematically evaluating how well different configurations detected anomalies.

As shown in Table 5, the MSE model applied to chroma CQT features achieved the best overall results, with an AUC of 0.7437, precision of 0.5795, recall of 0.8760, accuracy of 0.7118, and abnormal F1 of 0.7247. These results demonstrate that the MSE model with chroma CQT is highly effective for detecting anomalies in sound data, making it the best candidate for use as a baseline model. In comparison, the Huber model also performed well, especially in recall, reaching an impressive 0.9225 when applied to both chroma CQT and chroma CENS features. However, the slightly lower overall AUC and accuracy compared to the MSE model suggest that while Huber captures more anomalies (as seen in recall), it may not generalise as well across all metrics.

The MAE model showed moderate results. Its best performance was in the chroma CQT feature set, where it achieved an AUC of 0.7159 and an accuracy of 0.6735. Although its results are reasonable, they fall short of those of MSE and Huber. The chroma STFT features consistently showed lower performance across all models than chroma CQT and chroma CENS. The MSE model, for instance, achieved an AUC of 0.6842 with chroma STFT, which is notably lower than its performance with chroma CQT. However, the results also highlight limitations in the autoencoder’s ability to accurately reconstruct the input features, as illustrated in Figure 6. Even after 30 epochs of training, the reconstructed outputs diverged from the original inputs, suggesting that the model may not capture all the necessary patterns for robust anomaly detection. This is further supported by the loss distribution plots, where a significant overlap between normal and abnormal data was observed, making it difficult for the model to differentiate between them. Given the promising results from the MSE model with chroma CQT features, this configuration will serve as our baseline for future comparisons. We aim to refine and enhance the model by comparing it with the proposed CASSAD model, incorporating additional preprocessing techniques to improve detection accuracy and robustness in real-world applications.

### 5.2. Detection Results Using Isolation Forest

We also applied the isolation forest model to the same dataset used for the autoencoder, maintaining the same split for training and testing: 512 normal, 211 abnormal, and 129 normal samples for testing. We extracted the mean and standard deviation from the features and standardised the values using MinMax scaling between −1 and 1. The isolation forest model demonstrated strong performance across various chroma features for anomaly detection in idler components, as shown in Figure 7.

When applied to CQT features, the isolation forest algorithm achieved an AUC of 0.8568, with a recall of 0.7941, precision of 0.7926, and accuracy of 0.7941. This shows robust capabilities in distinguishing between normal and abnormal conditions, significantly outperforming the autoencoder, which had an AUC of 0.7437 and an accuracy of 0.7118 for the same features. Similarly, isolation forest’s performance on CENS features was notable, achieving an AUC of 0.8544 compared to the autoencoder’s lower performance of 0.7030. For STFT features, isolation forest achieved an AUC of 0.8576, slightly higher than the autoencoder’s result of 0.6842. One consistent finding across all feature sets was the stability of isolation forest’s results. Unlike the autoencoder, whose performance varied based on the feature set and loss function used, isolation forest delivered strong and consistent results across different chroma features with minimal variability.

The application of PCA with isolation forest generally led to a slight decrease in performance. For instance, when using CQT features, the AUC dropped from 0.8568 to 0.7423 after applying PCA. This suggests that while PCA helps with dimensionality reduction, it may obscure critical feature patterns needed for accurate anomaly detection, leading to reduced model performance. This trend was consistent across all chroma features. For STFT and CENS, PCA similarly reduced performance slightly, but the model’s ability to detect abnormalities remained competitive. The results highlight the significance of retaining important feature patterns that may be lost during PCA, especially in complex datasets like rotating machinery.

### 5.3. Detection Results Using LOF

Figure 8 highlights the LOF model’s strong performance across various feature sets, particularly with CENS features. For CENS features, the model achieved an AUC of 0.8395 when mean values were included and 0.8122 when only standard deviation values were used. This emphasises the importance of using mean aggregation to improve anomaly detection performance. This performance is comparable to the isolation forest model, which achieved an AUC of 0.8544 with mean values and 0.8177 without mean values for CENS features, suggesting that LOF and isolation forest are well-suited for anomaly detection using CENS features. The application of PCA further improved the LOF model’s performance, increasing the AUC to 0.8692 with mean values, showing that LOF benefits from both the inclusion of mean values and dimensionality reduction. However, the autoencoder model showed a slightly lower performance with CENS features, achieving a maximum AUC of 0.7083 when using the MSE loss function, indicating that LOF and isolation forest are more effective in handling this feature set.

For CQT features, LOF performed with an AUC of 0.7188 (with mean) and 0.7809 (without mean). After applying PCA, the AUC increased to 0.7390 with mean values and 0.8011 without mean values, indicating that PCA enhances the model’s ability to balance precision and recall, especially in the absence of mean values. This is consistent with the results from isolation forest, which also saw a slight drop in performance after PCA was applied to CQT features, where the AUC decreased from 0.8568 (without PCA) to 0.7423 (with PCA) for mean values. The autoencoder, however, achieved an AUC of 0.7437 with CQT features using MSE, demonstrating slightly better performance than LOF when mean values were considered but lower than LOF’s performance without mean values. When compared to the LOF model and isolation forest, both performed well across various feature sets, with LOF showing greater flexibility when PCA was applied. In contrast, isolation forest generally performed better without PCA. Meanwhile, while competitive in some instances, the autoencoder model does not consistently match the performance of LOF or isolation forest.

### 5.4. Detection Results Using the Proposed Model (CASSAD)

The detection results using the proposed CASSAD model demonstrate significant improvements across various feature sets compared to traditional models such as one-class SVM, the local outlier factor (LOF), isolation forest, and autoencoder. As shown in Table 6, the proposed CASSAD model consistently achieves higher performance metrics regarding AUC, recall, precision, abnormal (−1) F1, and accuracy. This superiority is particularly evident with high-dimensional features like “All Chroma Features”, where the proposed model outperforms all other models. For “All Chroma Features” with mean values, the proposed CASSAD model achieves an AUC of 0.96, significantly improving over the previous one-class SVM results. Even when employing PCA, the model maintains a high AUC of 0.94, indicating its robustness in managing high-dimensional data while preserving essential information. These results highlight the model’s capability to capture complex datasets’ characteristics effectively. When examining “Std” features, the proposed CASSAD model delivers superior results with an AUC of 0.83 without PCA and 0.73 with PCA. These metrics suggest the model is highly effective across different feature processing approaches, confirming its adaptability in detecting anomalies within idler components.

Overall, the proposed CASSAD model demonstrates proficiency in handling temporal frequency and high-dimensional data patterns, making it a suitable choice for anomaly detection in industrial applications. It consistently outperforms LOF, isolation forest, and autoencoder models, which show comparatively lower performance metrics, especially when dealing with the intricacies of chroma features.

### 5.5. Visual Analysis of the Anomaly Detection Model

To directly present the diagnostic outcomes of the proposed CASSAD model, compared to the best results from LOF and Isolation Forest, we employ visual tools such as the confusion matrix, precision–recall curve, and t-distributed Stochastic Neighbor Embedding (t-SNE).

As illustrated in Figure 9, we employed a set of visual tools to evaluate the performance of various anomaly detection models. In the top row, the confusion matrices provide a detailed breakdown of the classification results for each model, highlighting the distribution of true positives, false positives, true negatives, and false negatives [50]. This helps us evaluate the model’s accuracy in distinguishing between normal and abnormal conditions. The bottom row presents ROC curves, precision–recall curves, and t-SNE visualisations. The ROC and precision–recall curves demonstrate the trade-offs between sensitivity and specificity, which are particularly useful for imbalanced datasets [51]. The t-SNE plots, on the other hand, visualise the class separability, showcasing how well each model distinguishes between normal and abnormal operational states. Together, these visualisations offer a robust model for assessing the performance of the models under study [52].

### 5.6. Effect of Noise Filtering

Noise filtering has significantly improved model performance in acoustic-based anomaly detection for rotating machinery components. This analysis evaluated the impact of noise filtering on various machine learning models, including the proposed CASSAD model, using OCSVM, focusing on two critical metrics: AUC (area under the curve) and accuracy. The comparison was done across several feature sets using the combined all chroma features dataset. The results were assessed before and after applying noise filtering to understand its influence on the model’s ability to detect anomalies.

Filtering and smoothing noise are essential steps in achieving high-performance metrics, as they help isolate meaningful patterns in the data by reducing ambient noise interference. Evaluating the proposed model’s effectiveness by measuring performance before and after noise filtering provides a clearer understanding of the contribution to model improvements. The findings indicate that noise filtering consistently improves model performance across all feature sets, with significant gains in AUC and accuracy. For example, with the CASSAD model applied to the all chroma features dataset, the AUC increased from 0.8529 to 0.9629 after noise filtering, and the accuracy improved from 0.7759 to 0.9059. This demonstrates the value of combining noise filtering with feature enhancement techniques to boost the robustness of anomaly detection models.

The comparative plot in Figure 10 clearly highlights the improvements in both AUC and accuracy for the proposed model following the application of noise filtering. The bar plots clearly illustrate how the model benefits from this preprocessing technique. Noise filtering, combined with mean computation, proves essential for enhancing the precision, reliability, and overall performance of anomaly detection models in noisy environments.

### 5.7. Training and Evaluation Time Analysis

The training and evaluation times of LOF, isolation forest, and the proposed model were analysed across feature extraction methods (CQT, STFT, CENS, and all chroma features) with and without PCA. The proposed model consistently demonstrated lower training times than LOF and isolation forest, highlighting its computational efficiency. This is illustrated in Figure 11.

All models performed well for evaluation, with times generally remaining below 0.1 s, suggesting that the evaluation phase is less sensitive to model complexity and more reflective of the number of data points being processed. Applying PCA further reduced training and evaluation times, with isolation forest showing the most significant reduction, particularly when using all chroma features. As feature complexity increased, the computational cost rose for training in all models, reflecting a trade-off between feature richness and efficiency. These results highlight the proposed model’s suitability for real-time applications due to its faster training times, while all models demonstrated efficient evaluation performance.

### 5.8. Comparison with Previous Work

In this study, we evaluate the proposed CASSAD model compared to adaptations of the YAMNet model adjusted for binary classification tasks [5], utilising the same subset of 255 audio samples originally used for YAMNet. YAMNet, initially designed for multiclass tasks, has been modified using advanced neural network techniques such as bidirectional long short-term memory (BiLSTM) and bidirectional gated recurrent unit (BiGRU) networks to enhance its effectiveness in binary detection, which is crucial for detecting anomalies in idlers.

Our analysis focuses on identifying normal and abnormal sounds within the dataset, which consists of 101 normal and 154 abnormal samples. After data augmentation, the training set included 200 normal samples to help the model better recognise standard sound patterns. The test set consisted of 51 normal and 154 abnormal samples, providing a comprehensive performance evaluation.

The proposed CASSAD model achieved an AUC of 0.93 and an accuracy of 93.25% using all chroma features, setting a new benchmark for comparison against the adapted YAMNet models. As shown in Table 7, the proposed CASSAD model exceeds the YAMNet adaptations in accuracy and demonstrates superior performance across other key metrics such as recall and F1 score. This highlights the model’s robustness in recognising complex patterns in audio data, with the significant contribution from all chroma features emphasising their importance in effectively distinguishing between normal and abnormal sounds.

While the YAMNet adaptations show promise, their need for extensive computational resources and preprocessing could limit their practical application in real-time scenarios. The proposed model offers a more efficient solution, achieving high performance without the same computational overhead. This comparative analysis advances the discussion on employing binary detection for effective anomaly detection, particularly in contexts where multiclass models struggle due to class imbalances or the rarity of anomalies.

### 5.9. Discussion

This study evaluated four different anomaly detection models: autoencoder, LOF, iForest, and the proposed CASSAD model using OCSVM. Each method has its advantages and limitations when detecting faults in machinery sound data.

Compared to traditional models such as autoencoder, LOF, and iForest, the CASSAD model demonstrated superior performance in identifying faults in idler sound data. As a neural network-based approach, the autoencoder learns to reconstruct normal data and detects anomalies through reconstruction errors. While it excels at modelling complex patterns in high-dimensional data, its performance is highly dependent on hyperparameter settings and requires substantial training data for effective anomaly detection [53].

LOF assesses the isolation of data points by comparing their local density to that of their neighbours, which makes it effective in detecting anomalies within clustered data. However, its computational complexity and sensitivity to parameter tuning, such as the neighbourhood size, can pose challenges, particularly when dealing with large datasets. iForest identifies anomalies by randomly partitioning the data, with points with shorter paths considered potential outliers. Although efficient for large datasets and free from assumptions about data distribution, its performance can vary significantly depending on parameter settings, such as the number of trees [31].

The CASSAD model has shown strong performance in detecting localised faults by effectively identifying anomalies in specific regions of the sound data. However, capturing global patterns remains challenging, particularly in noisy environments where overlapping noise can obscure broader anomalies. While preprocessing steps like noise filtering help mitigate some interference, the model’s current emphasis on local patterns may limit its ability to detect global anomalies, especially in high-dimensional datasets. Future studies need to look into incorporating more sophisticated strategies, like variational autoencoders (VAE) [53] or transformer-based models [54]. These are better prepared to understand the base data distribution and provide a broader approach for detecting anomalies. Such models can boost the model’s capability to distinguish between local and global irregularities. This way, they increase the precision of fault detection in complicated conditions with much noise.

## 6. Conclusions and Future Work

This study proposes the chroma-augmented semi-supervised anomaly detection (CASSAD) model for identifying anomalies in industrial machinery, specifically in conveyor belt idler components. By leveraging OSVM and chromagram audio features such as CENS, CQT, and STFT, the CASSAD model effectively addresses the challenge of anomaly detection in noisy environments with limited labelled data. Integrating mean values with OSVM yielded the highest accuracy and AUC scores, demonstrating the potential of unsupervised approaches for one-class industrial fault detection tasks. Incorporating feature filtering and augmentation with AWGN further improved performance, highlighting the significance of robust preprocessing techniques. These findings indicate that the CASSAD model offers a practical and scalable solution for anomaly detection in real-world industrial settings, even without extensive labelled data. Future work will focus on integrating the CASSAD model into real-time monitoring systems to enable continuous and automated fault detection. Additionally, exploring advanced semi-supervised learning techniques, such as pseudo-labelling and self-training, will allow for incorporating stage labels to enhance decision boundaries. Efforts will also be directed towards expanding the dataset to include diverse operational conditions and conducting comprehensive evaluations, such as learning curve analyses and cross-validation consistency checks, to ensure the model’s adaptability and reliability. This research provides a foundation for developing scalable, efficient, and reliable anomaly detection systems for industrial applications, especially in conveyor belt systems where minimising downtime and improving operational efficiency is crucial.

## Figures and Tables

**Figure 1 sensors-24-07569-f001:**
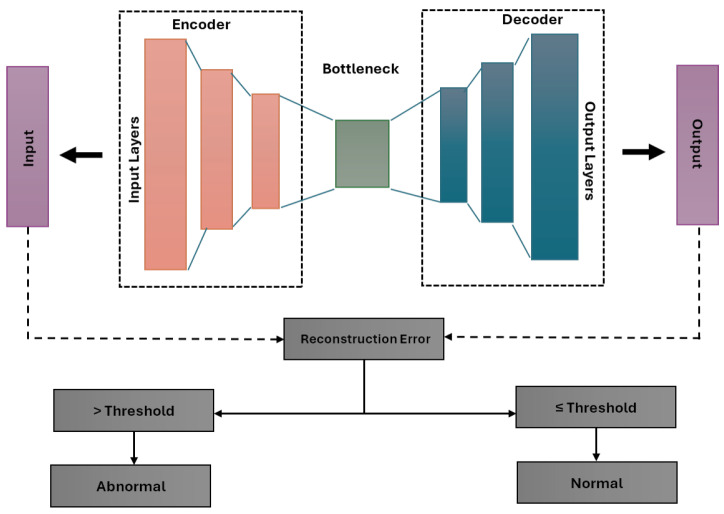
Autoencoder structure.

**Figure 2 sensors-24-07569-f002:**
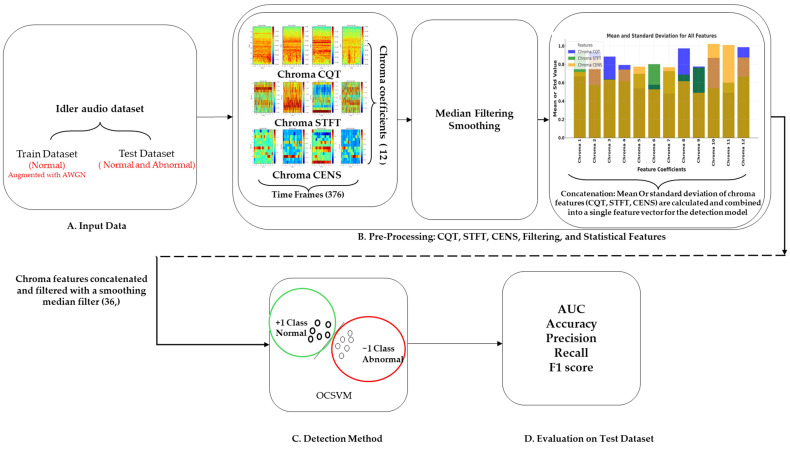
Overview of the proposed chroma-augmented semi-supervised anomaly detection (CASSAD) model.

**Figure 3 sensors-24-07569-f003:**
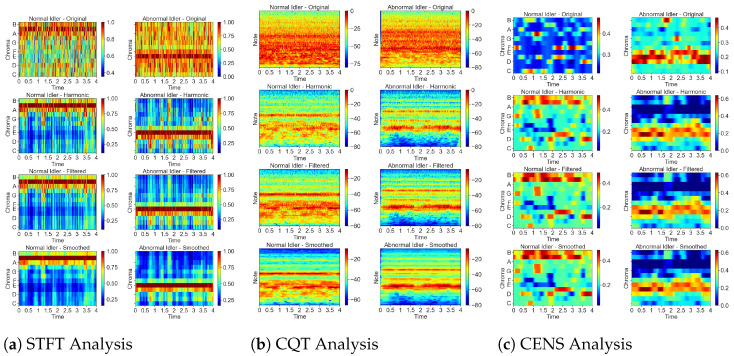
Analysis of different chroma features across four stages: original, harmonic, filtered, and smoothed, showing (**a**) STFT, (**b**) CQT, and (**c**) CENS.

**Figure 4 sensors-24-07569-f004:**
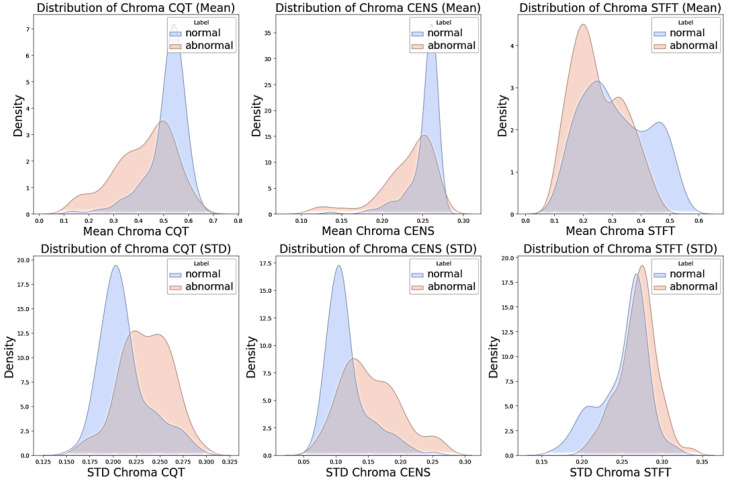
Distribution of mean and standard deviation (STD) of chroma features (CQT, CENS, and STFT) for normal and abnormal signals.

**Figure 5 sensors-24-07569-f005:**
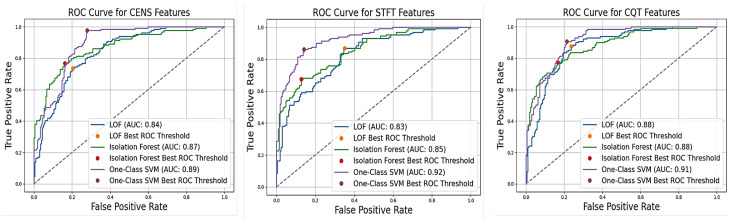
ROC curves with the best AUC for LOF, isolation forest, and one-class SVM models.

**Figure 6 sensors-24-07569-f006:**
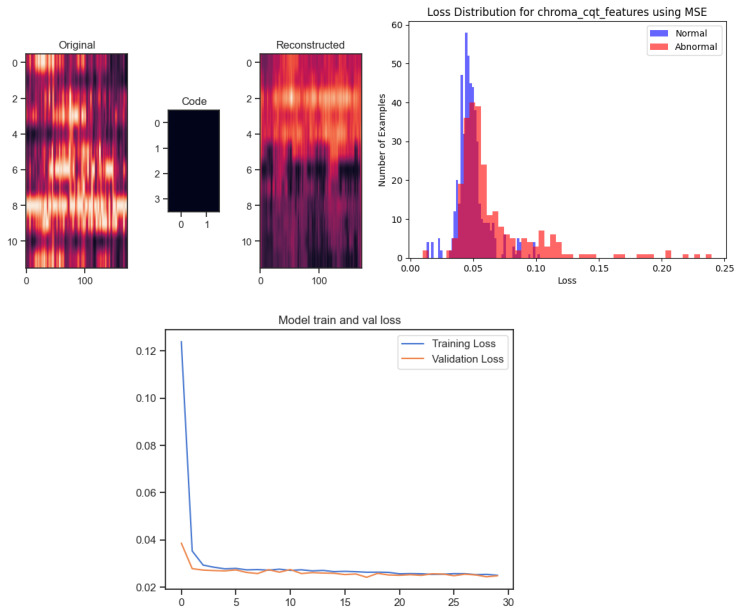
Analysis results from training the autoencoder on chroma features. (**Top Row**): Original and reconstructed spectrograms (**Left**) and the loss distribution for chroma_cens_features using the mean absolute error (MAE) function (**Right**). (**Bottom row**): Training and validation loss curves over 30 epochs showing minimal gap and no significant overfitting or underfitting.

**Figure 7 sensors-24-07569-f007:**
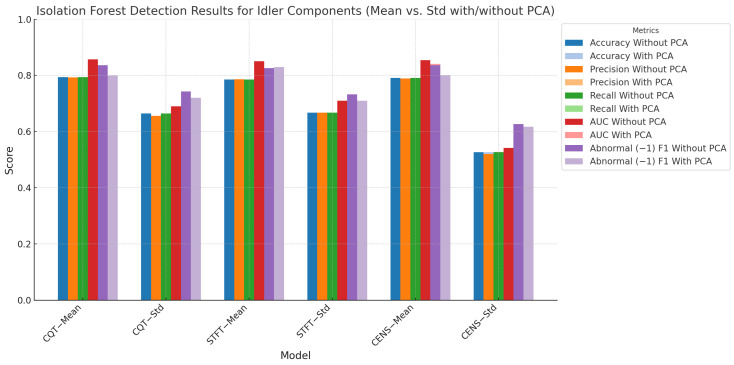
Isolation forest detection results for idler components (mean vs. standard deviation with/without PCA). This figure highlights the model’s performance across different chroma features and aggregation methods.

**Figure 8 sensors-24-07569-f008:**
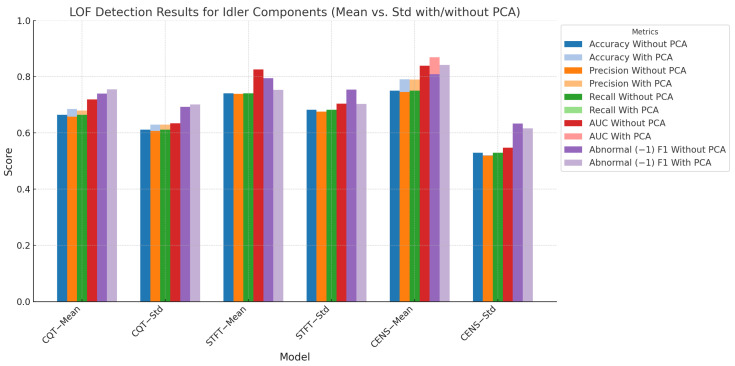
LOF detection results for idler components (mean vs. standard deviation with/without PCA). This figure highlights the model’s performance across different chroma features and aggregation methods.

**Figure 9 sensors-24-07569-f009:**
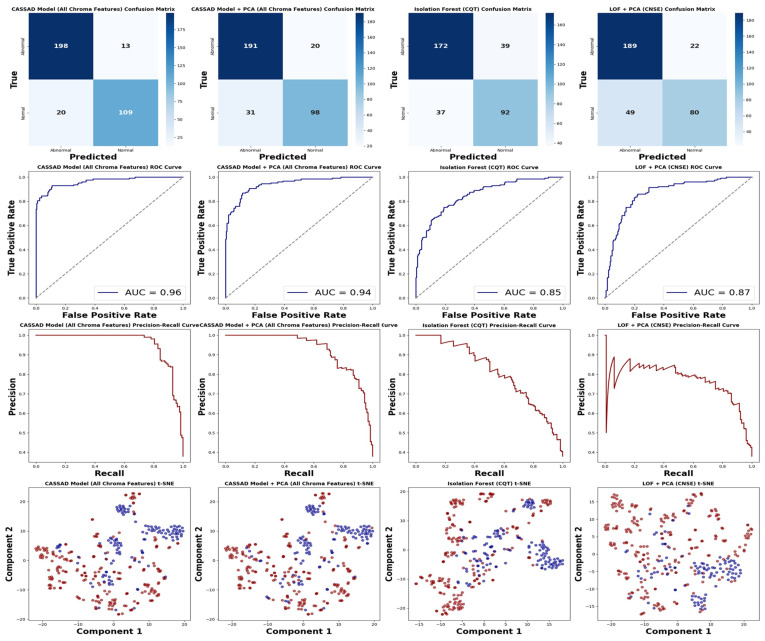
Visualisation of the best results for LOF + PCA, isolation forest, and the proposed CASSAD model. (**Top row**): Confusion matrices showing classification results. (**Second row**): ROC curves, with the proposed CASSAD model reaching the highest AUC of 0.96. (**Third row**): Precision–recall curves, where the proposed CASSAD model without PCA shows the best balance. (**Bottom row**): t-SNE plots illustrating data separability, with the proposed CASSAD model achieving the clearest distinction.

**Figure 10 sensors-24-07569-f010:**
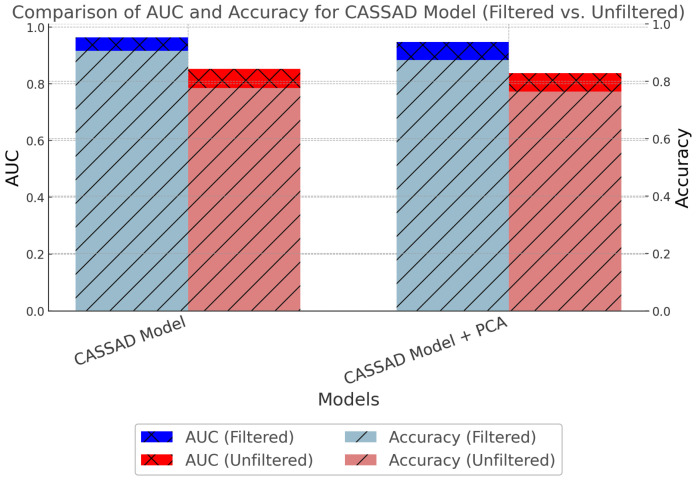
Comparison of AUC and accuracy metrics for the proposed model using all chroma features, both with and without noise filtering. The figure highlights the performance improvements achieved by the proposed model (CASSAD).

**Figure 11 sensors-24-07569-f011:**
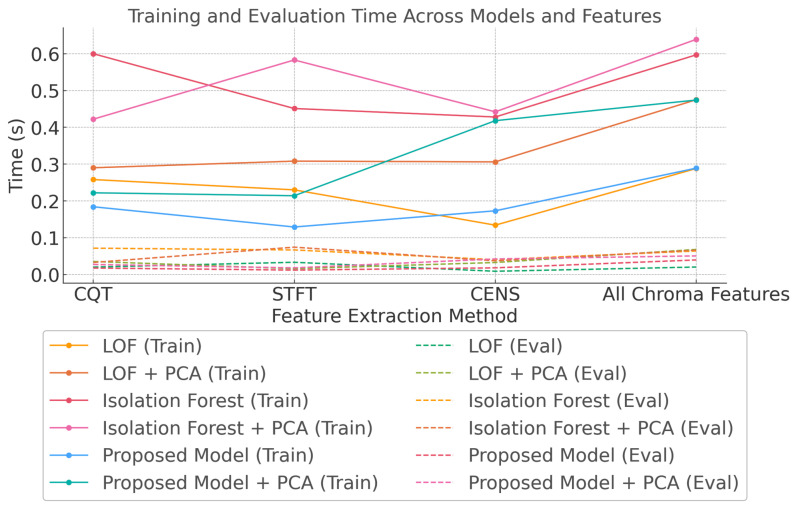
Consumption time comparison across models and features.

**Table 1 sensors-24-07569-t001:** A comparison of the anomaly detection models explored.

Model	Key Concept	Advantages	Drawbacks
**OCSVM**	Creates a boundary around normal data points in high-dimensional space to differentiate them from anomalies [34].	Works well with high-dimensional data and can model complex boundaries.	Requires careful tuning of parameters like the kernel type and regularisation. It can also be sensitive to noise.
**LOF**	Assesses the local density of a data point compared to its neighbours to detect anomalies [32].	Effectively captures the local data structure and is great for detecting anomalies in clustered data.	Can be computationally demanding for large datasets and requires careful neighbourhood size selection.
**iForest**	Detects anomalies by recursively partitioning the data with random splits, leading to shorter paths for anomalies [35].	Efficient for large datasets and does not rely on any assumptions about data distribution.	Its performance can vary based on the random splits and is sensitive to settings like the number of trees.

**Table 2 sensors-24-07569-t002:** Number of samples by condition.

Condition	Description	Files
Normal	Baseline measurements or standard operation conditions of idlers.	257
Abnormal	Includes stage 1, 2, and 3 faults in bearings, indicating deviations from normal conditions.	211

**Table 3 sensors-24-07569-t003:** Hyperparameters used for each model in our experiments.

Model	Parameter	Value
Autoencoder	Encoder Layers	128, 64, 32, 16
	Decoder Layers	32, 64, 128
	Dropout	0.2
	Activation	ReLU
	Loss Functions	MAE, MSE, Huber
	Epochs	30
	Batch Size	4
iForest	Contamination	0.01
LOF	Contamination	0.01
	Novelty	True
OCSVM	Nu	0.005
	Gamma	1
	Kernel	RBF

**Table 4 sensors-24-07569-t004:** Best ROC thresholds and AUC for LOF, isolation forest, and one-class SVM models across different feature sets.

Features	Model	Best ROC Threshold	AUC
CQT	LOF	0.59	0.876
CQT	Isolation Forest	0.24	0.883
CQT	One-Class SVM	−0.02	0.912
STFT	LOF	1.26	0.825
STFT	Isolation Forest	0.10	0.829
STFT	One-Class SVM	−0.02	0.924
CENS	LOF	0.83	0.840
CENS	Isolation Forest	0.20	0.867
CENS	One-Class SVM	-0.04	0.891

**Table 5 sensors-24-07569-t005:** Autoencoder detection results with the best metrics highlighted.

#	Model	Chroma Features	Accuracy	Precision	Recall	Abnormal (−1) F1	AUC
0	MAE	chroma CQT	0.6735	0.5425	0.8915	0.6726	0.7159
1	MSE	chroma CQT	**0.7118**	**0.5795**	0.8760	**0.7247**	**0.7437**
2	Huber	chroma CQT	0.6882	0.5535	0.9225	0.6845	0.7338
3	MAE	chroma CENS	0.6441	0.5174	0.9225	0.6231	0.6982
4	MSE	chroma CENS	0.6529	0.5238	0.9380	0.6313	0.7083
5	Huber	chroma CENS	0.6500	0.5219	0.9225	0.6316	0.7030
6	MAE	chroma STFT	0.6118	0.4928	0.7907	0.6163	0.6465
7	MSE	chroma STFT	0.6735	0.5529	0.7287	0.7087	0.6842
8	Huber	chroma STFT	0.5912	0.4777	0.8295	0.5749	0.6375

**Note:** Bolded values indicate the best-performing metrics for each evaluation criterion.

**Table 6 sensors-24-07569-t006:** Detection results for idle components using CASSAD with one-class SVM, comparing mean and standard deviation with/without PCA.

Chroma Features	Aggregation	Model	Accuracy	Precision	Recall	Abnormal (−1) F1	AUC
**CQT**	Mean	One-Class SVM	0.6971	0.6949	0.6971	0.7588	0.7571
		One-Class SVM + PCA	0.6794	0.6692	0.6794	0.7604	0.7536
	Std	One-Class SVM	0.5941	0.5881	0.5941	0.6806	0.6331
		One-Class SVM + PCA	0.6029	0.5922	0.6029	0.6939	0.6485
**STFT**	Mean	One-Class SVM	0.8441	0.8448	0.8441	0.8804	0.9241
		One-Class SVM + PCA	0.7912	0.7888	0.7912	0.8368	0.8649
	Std	One-Class SVM	0.7324	0.7288	0.7324	0.7898	0.7385
		One-Class SVM + PCA	0.6676	0.6612	0.6676	0.7414	0.7034
**CENS**	Mean	One-Class SVM	0.7500	0.7459	0.7500	0.8098	0.8906
		One-Class SVM + PCA	0.7706	0.7675	0.7706	0.8211	0.8725
	Std	One-Class SVM	0.5176	0.5205	0.5176	0.6077	0.5363
		One-Class SVM + PCA	0.5294	0.5308	0.5294	0.6190	0.5469
**All Features**	Mean	**Proposed Model**	**0.9059**	**0.9059**	**0.9059**	**0.9242**	**0.9629**
		**Proposed Model + PCA**	**0.8735**	**0.8731**	**0.8735**	**0.8988**	**0.9474**
	Std	Proposed Model	0.8000	0.8025	0.8000	0.8150	0.8350
		Proposed Model + PCA	0.6941	0.6872	0.6941	0.7647	0.7362

**Note:** Bolded values represent the best performance across all models and feature aggregation techniques.

**Table 7 sensors-24-07569-t007:** Performance metrics for machine learning models handling audio classification in a binary format. Abbreviations: Acc. = accuracy, Prec. = precision, Rec. = recall, and F1 = F1 score.

Model Name	Acc. (%)	Prec. (%)	Rec. (%)	F1 (%)
YAMNet BiLSTM with Attention	91.59	94.56	90.29	92.37
YAMNet BiGRU with Attention	91.59	94.07	91.00	92.51
YAMNet with LazyAdam Optimiser	92.18	94.25	91.79	93.00
**Proposed Model (CASSAD)**	**93.00**	94.50	**92.00**	**93.25**

**Note:** The best performance for each metric is highlighted in bold.

## Data Availability

Data will be made available on request.

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
