# Peer review of "CASSAD: Chroma-Augmented Semi-Supervised Anomaly Detection for Conveyor Belt Idlers"

_sensors, 2024, doi:10.3390/s24237569_

Round 1
Reviewer 1 Report
Comments and Suggestions for Authors
This manuscript proposes a Chroma-Augmented Semi-Supervised Anomaly Detection (CASSAD) method by utilizing the mean and standard deviation across chroma features. The dataset is further augmented using Additive White Gaussian Noise. The proposed method can perform effectively with limited labelled data. It’s meaningful and interesting. However, some important issues are not introduced clear. I don’t think it suitable for publication in Sensors at current version.
1. What is the chroma-augmented semi-supervised anomaly detection theory? Which sensor is used? Where the sensor is located? The necessary detection system hardware must be illustrated. The necessary detection mathematical model must be established.
2. The autoencoder structure as shown in Figure 1 is too simple to be understand. More details must be illustrated in Figure 1.
3.How to get the reachability distance in Eq.(3)? What’s the physical meaning of (x, y)?
4.There are many parameters are not introduced in advance. The readability is challenging.
5.How to verify the number of samples is good enough?
6. Accuracy is certainly important, but time consumption is also a reflection efficiency indicator that must be assessed.
Author Response
Responses to Reviewer 1
Comment 1
- What is the chroma-augmented semi-supervised anomaly detection theory? Which sensor is used? Where is the sensor located? The necessary detection system hardware must be illustrated. The necessary detection mathematical model must be established.
Response from the authors:
Thank you for your comment. The proposed approach utilizes chroma features, specifically CQT, STFT, and CENS, to capture the harmonic characteristics of audio signals. These features are augmented and processed in a semi-supervised framework using One-Class SVM, which is trained exclusively on normal data. The method identifies anomalies by detecting deviations from the learned normal patterns, ensuring efficient and accurate acoustic anomaly detection in industrial environments.
The detection system relies on high-quality acoustic sensors (microphones) strategically placed near rotating belt conveyor idlers' rotating components to capture high-relevant and high-quality sound signals. The dataset section provides details regarding the sensors and their placement. This study focuses on developing a robust and efficient anomaly detection method rather than proposing a fully integrated hardware-based detection system. However, future work will aim to design a comprehensive detection system, incorporating components such as signal processing units, edge computing devices, and communication modules to support real-time deployment.
The Background Section thoroughly explains the mathematical foundation of the proposed detection method based on One-Class SVM. Future research will extend this model to include system-level constraints and optimizations as part of a broader hardware-integrated anomaly detection framework.
Comment 2
- The autoencoder structure as shown in Figure 1 is too simple to be understand. More details must be illustrated in Figure 1.
Response from the authors:
Thank you for your feedback. We have updated Figure 1 with additional details for clarity. Please see the revised version below.

Comment 3
3.How to get the reachability distance in Eq.(3)? What’s the physical meaning of (x, y)?
Response from the authors:
Thank you for your question. The reachability distance in Eq. (3) is calculated as described in the updated text below. Additionally, the physical meaning of (x,y) has been clarified in the revised explanation. Please refer to the updated text in red color for more details.

Comment 4
4.There are many parameters are not introduced in advance. The readability is challenging.
Response from the authors:
Thank you for your observation regarding parameter introduction. To address this concern, we have ensured that all parameters used in our methodology are clearly listed and explained in Table 3, including their respective configurations. Additionally, we have reviewed the manuscript thoroughly to verify that all parameters critical to understanding our work are introduced in the text before their use in any tables or figures.
We would greatly appreciate it if the reviewer could specify any particular parameters or aspects that remain unclear or are not adequately introduced. This additional feedback would help us further improve the clarity and readability of the manuscript.
Comment 5
5.How to verify the number of samples is good enough?
Response from the authors:
Thank you for your valuable question. To verify whether the number of samples is sufficient, we have designed the dataset to capture diverse operational scenarios, including normal operation and multiple fault stages, informed by domain knowledge to reflect realistic conditions. Additionally, in future work, we plan to expand the dataset and conduct rigorous analyses such as learning curve evaluations and cross-validation consistency checks. These steps will help ensure the adequacy of the sample size and assess the model's robustness across various conditions. Please see the updated text in red for more details.

Comment 6
- Accuracy is certainly important, but time consumption is also a reflection efficiency indicator that must be assessed.
Response from the authors:
Thank you for the feedback. We agree that time consumption is a key efficiency indicator. See below the updated analysis.

Reviewer 2 Report
Comments and Suggestions for Authors
The authors mente CASSAD, that supersedes its predecessors in that it allows incalculably fewer labelled examples for its application. They added chroma features and used One-Class SVM as the main model and carried out results comparisons with other models such as LOF and iForest. The research further took out the CASSAD approach on a particular existing dataset of sounds of idlers being used in industries, where it performed better in detecting failures, with an AUROC of 96% and accuracy of 91%.
Combine section 1 and 2.1.
The novelty of the paper is not clear
Figure 2 is not readible. Please arrange it again
Tables 6 and 7 can give as Figure
Conclusion section can be given as bullets
Author Response
Responses to Reviewer 2
Comment 1
Combine section 1 and 2.1.
Response from the authors:
Thank you for the suggestion. Sections 1 and 2.1 have been combined for clarity. Please see below.

Comment 2
The novelty of the paper is not clear
Response from the authors:
Thank you for your feedback. The introduction has been updated to make the novelty of the paper clearer. Please see the revised section for details.

Comment 3
Figure 2 is not readable. Please arrange it again
Response from the authors:
Thank you for your feedback. The figure has been updated for better readability. Please see the revised version below.

Comment 4
Tables 6 and 7 can give as Figure
Response from the authors:
Thank you for your suggestion. Tables 6 and 7 have been updated and presented as figures for better visualization. Please see the revised figures below.

Comment 5
Conclusion section can be given as bullets
Response from the authors:
Thank you for your suggestion. However, we believe the conclusion section is most effective as a coherent narrative rather than bullet points. This format ensures a smoother flow and helps comprehensively connect the main findings, contributions, and implications. We have carefully reviewed the section and made improvements to enhance clarity and conciseness while retaining the narrative style. We hope this addresses your concern.

Round 2
Reviewer 1 Report
Comments and Suggestions for Authors
This manuscript proposes a Chroma-Augmented Semi-Supervised Anomaly Detection (CASSAD) method by utilizing the mean and standard deviation across chroma features. The dataset is further augmented using Additive White Gaussian Noise. The proposed method can perform effectively with limited labelled data. It’s meaningful and interesting. The authors have revised this manuscript according to the comments. I think it suitable for publication in Sensors now.